# Untargeted LC-MS/MS Metabolomics Study of HO-AAVPA and VPA on Breast Cancer Cell Lines

**DOI:** 10.3390/ijms241914543

**Published:** 2023-09-26

**Authors:** Alan Rubén Estrada-Pérez, Juan Benjamín García-Vázquez, Humberto L. Mendoza-Figueroa, Martha Cecilia Rosales-Hernández, Cynthia Fernández-Pomares, José Correa-Basurto

**Affiliations:** 1Laboratorio de Diseño y Desarrollo de Nuevos Fármacos e Innovación Biotecnológica, Escuela Superior de Medicina, Instituto Politécnico Nacional, Plan de San Luis y Díaz Mirón s/n, Casco de Santo Tomás, Ciudad de México 11340, Mexico; 2Laboratorio de Biofísica y Biocatálisis, Sección de Estudios de Posgrado e Investigación, Escuela Superior de Medicina, Instituto Politécnico Nacional, Plan de San Luis y Díaz Mirón s/n, Casco de Santo Tomás, Ciudad de México 11340, Mexico

**Keywords:** metabolomics, breast cancer, valproic acid, HO-AAVPA, MDA-MB-231, MCF-7

## Abstract

Breast cancer (BC) is one of the biggest health problems worldwide, characterized by intricate metabolic and biochemical complexities stemming from pronounced variations across dysregulated molecular pathways. If BC is not diagnosed early, complications may lead to death. Thus, the pursuit of novel therapeutic avenues persists, notably focusing on epigenetic pathways such as histone deacetylases (HDACs). The compound *N*-(2-hydroxyphenyl)-2-propylpentanamide (HO-AAVPA), a derivative of valproic acid (VPA), has emerged as a promising candidate warranting pre-clinical investigation. HO-AAVPA is an HDAC inhibitor with antiproliferative effects on BC, but its molecular mechanism has yet to be deciphered. Furthermore, in the present study, we determined the metabolomic effects of HO-AAVPA and VPA on cells of luminal breast cancer (MCF-7) and triple-negative breast cancer (MDA-MB-231) subtypes. The LC-MS untargeted metabolomic study allowed for the simultaneous measurement of multiple metabolites and pathways, identifying that both compounds affect glycerophospholipid and sphingolipid metabolism in the MCF-7 and MDA-MB-231 cell lines, suggesting that other biological targets were different from HDACs. In addition, there are different dysregulate metabolites, possibly due to the physicochemical differences between HO-AAVPA and VPA.

## 1. Introduction

Breast cancer (BC) has the highest incidence and is the leading cause of death due to cancer worldwide [1]. Early BC detection usually results in a favorable prognosis; however, if early detection fails, invasive BC treatment options are ineffective [1,2,3]. One of the biggest challenges for BC treatment is its heterogeneity, exemplified by BC subtypes that show a high diversity of genetic and epigenetic origins, which determine its classification and treatment and are directly related to the state of relevant receptors such as estrogen receptor (ER), progesterone receptor (PR), and human epithelial receptor 2 (HER2) [2]. Differences in the expression of these receptors have been used to indicate prognosis; the latter ranges from BC luminal A (LA) to triple-negative BC (TNBC), with highly contrasting prognoses: better prognosis and lower aggressiveness for LA and worse prognosis and high aggressiveness for TNBC [2].

The role of epigenetic modifications in carcinogenesis and cancer development has become more relevant due to environmental factors [4]. In particular, histone acetylation and deacetylation by histone acetyltransferases (HAT) and histone deacetylases (HDAC), respectively, have gained great importance due to their involvement in the regulation of transcription processes for genes implicated in carcinogenesis and cancer development [5,6]. In normal conditions, HDACs maintain histones positively charged, favoring their interactions with DNA and regulating the expression of genes involved in tumorigenesis. However, in cancer cells, HDAC overexpression disrupts gene regulation, causing the expression of genes associated with carcinogenesis and cancer development [5,7]. In this regard, the deregulation of HDAC has been identified as one of the main components of the origin of breast cancer, and thus, these proteins are considered high-value targets for its treatment [6,8,9,10,11]. Then, special attention has been given to HDAC inhibitors (HDACi) due to their capacity to restore altered epigenetic pathways, enabling the correct function of mechanisms that deal with cancer cells [7,12,13,14,15,16,17,18,19,20]. One of these compounds is valproic acid (VPA), an HDACi that acts on classes I and IIa HDACs [7,21]. In vitro studies have demonstrated that VPA can induce cell cycle arrest and apoptosis and affect cell migration in multiple breast cancer cell lines [22,23,24,25]. Similar to suberoylanilide hydroxamic acid (SAHA) and other HDACi, VPA also regulates cellular metabolism in cancer cells [26,27,28,29]. This metabolic regulation occurs independently of the transcriptional regulation of specific metabolism-related genes, a mechanism that also contributes to its anticancer activity [26].

Despite its therapeutic effects, the use of VPA is limited by its high concentrations, which can lead to toxicity. As a result, several VPA-derived molecules have been proposed; the most notable is compound *N*-(2-hydroxyphenyl)-2-propylpentanamide (HO-AAVPA), which is a VPA derivative [30,31,32,33,34]. HO-AAVPA exerts similar effects to VPA in different experimental contexts. For example, HO-AAVPA possesses an anticonvulsant effect comparable to VPA but with lower toxicity and teratogenic effects in an in vivo model [35]. Additionally, HO-AAVPA modifies HMGB1 acetylation, a nonhistone protein involved in DNA stability, repair, transcription, and recombination processes and in the generation of reactive oxygen species (ROS), activities which are also found in VPA with lower potency [36]. Although HO-AAVPA was designed as an HDACi against HDAC8 [37] based on docking affinity measurement, in vitro experiments have shown that HO-AAVPA has a higher affinity for HDAC1 [32]. In their original publication, Prestegui-Martlel et al. (2016) demonstrated that HO-AAVPA has antiproliferative activity against BC cells at lower concentrations compared to VPA [37]. Accumulating evidence supports the similar effects produced by HO-AAVPA and VPA, as well as the discrepancies between the molecular target for which it was designed and the findings obtained in vitro [35,36,37]. Furthermore, many pharmacological properties of HO-AAVPA remain to be elucidated, particularly its impact on intracellular metabolism of distinct subtypes of BC cells, such as LA and TNBC, two cell lines with varying levels of aggressiveness and distinct biological targets, including HDACs [9] and metabolomic profiles [38].

Due to the large magnitude and impact of epigenetic modifications regulated by HDACs, powerful tools are currently used to measure a large set of molecules in a reduced number of experiments. One example of such a tool is metabolomics, which uses experimental and bioinformatic methods to study metabolites. Unlike targeted metabolomics, the non-targeted approach allows for the simultaneous detection and quantification of thousands of small molecules, a global metabolic profile of the analyzed samples [39,40,41]. Through this strategy, it becomes possible to compare metabolic changes resulting from a particular treatment. This helps enhance comprehension of the biological processes and the metabolic pathways that are implicated. Although untargeted metabolomics is a valuable technique, it has some challenges. Signals do not represent metabolites directly, and they are considered “features” that have a specific *m*/*z* value, abundance, and retention time. Features are then processed to determine which of them are isotopes from adducts that belong to the same molecule, called “compound”. One reliable way to determine the identity of these compounds is by comparing the fragmentation spectra of specific “compounds” to those in metabolite libraries like the Human Metabolome Database (HMDB). If the two spectra match, the metabolite can be given a putative identity (metabolite identification confidence level 2) [42].

Under this approach, the present study aims to explore the metabolic changes induced by HO-AAVPA in two subtypes of breast cancer cells, contrasting with the effects produced by VPA. We hypothesize that HO-AAVPA impacts deregulated metabolic pathways in BC cells similar to VPA [43,44,45], but their physicochemical difference properties could yield different metabolomics profiles. The results of this work can increase the amount of knowledge about the effects elicited in cellular metabolites as well as contribute new evidence about the metabolic pathways that, when restored, induce the activation of the mechanisms involved in the elimination of BC cells from different subtypes, LA, represented by MCF-7 (less aggressive BC), and TNBC, represented by MDA-MB-231 (more aggressive BC). The following untargeted metabolomic study was carried out in two steps. Firstly, LC-MS was used to detect any features that were dysregulated by the treatments. Subsequently, LC-MS/MS was employed to gather fragmentation information to annotate the corresponding features as putative metabolites.

## 2. Results and Discussion

### 2.1. Inhibitory Concentration Assays

First, the effect on cell viability was determined for each compound in both cell lines (Figure 1). Then, the IC_15_ values were calculated from IC_50_ values (Table 1) and compared with those of previous reports. By observing the range of concentrations tested for both compounds (140 µM to 36 mM VPA and 9 µM to 2 mM HO-AAVPA), it is clear that HO-AAVPA has a higher potency, attributed to the 2-hydroxybenzamide fragment that functions as a zinc-binding group [35,36,37]. In their published study of HO-AAVPA, Prestegui-Martel et al. (2016) calculated IC_50_ values of 280 and 190 µM for MDA-MB-231 and MCF-7 cells, respectively [37]; according to our results, the calculated IC_50_ value was quite similar for MDA-MB-231 cells (291.8 µM) but not for MCF-7 cells (476.1 µM), although it was in the same order of magnitude. For VPA, a direct comparison was not possible because, in the aforementioned work, antiproliferative effects were not observed either in MDA-MB-231 or in MCF-7 cells at the maximum concentration tested; thus, authors determined that the effect was present at >450 and >350 µM for MDA-MB-231 and MCF-7 cells, respectively. However, Mawatari et al. (2015) found IC_50_ values for these cell lines on the same order of magnitude: 5.4 mM for MDA-MB-231 and 8.1 mM for MCF-7, compared with 7.29 and 7.11 mM in our study, respectively [25]. Other studies with different evaluation times have reported IC_50_ values ranging from 1.5 to 16 mM for MDA-MB-231 cells and from 0.7 to 8.1 mM for MCF-7 cells [24,46,47,48].

### 2.2. HO-AAVPA and VPA Effects on Breast Cancer Cells

Once IC_15_ values were obtained, the corresponding concentrations were applied to cell cultures to obtain the metabolomics samples. Through these experiments, 17 and 123 putative metabolites were found to be dysregulated by HO-AAVPA, and 12 and 20 were dysregulated by VPA in MDA-MB-231 (Table 2) and MCF-7 (Table 3) cells, respectively. For MDA-MB-231 cells, 10 putative metabolites were commonly affected by both compounds, although the trend (or direction) of dysregulation differed. Appendix A summarize the putative identification values for the deregulated entities in the MDA-MB-231 and MCF7 cell lines, respectively.

In MCF-7 cells, a higher number of shared putative metabolites were affected by both compounds (15 compounds), and, contrary to what we found in MDA-MB-231 cells, 6 were dysregulated in the same direction (downregulated). MCF-7-treated cells showed a tendency toward downregulation; however, the observed trend for MDA-MB-231-treated cells was the opposite: metabolites from HO-AAVPA-treated cells were mainly downregulated, while metabolites from VPA-treated cells were predominantly upregulated.

### 2.3. Effects of HO-AAVPA and VPA on the Metabolic Pathways of Breast Cancer Cells

Determination of the main affected pathways was performed through Pathway Analysis from Metaboanalyst, which considers both the number of altered elements in a pathway with respect to the total and the position they occupy in the pathway to calculate the impact score (Figure 2); however, this analysis does not consider the fold change direction [49]. As a result of the pathway impact analysis, some of the pathways were found to be unimpactful and are reported with a low score value (even zero in some cases), which means that a low number of members of the corresponding pathway (it might be only one member) was found in the dataset analyzed. Although MCF-7 cells treated with HO-AAVPA showed the highest number of dysregulated metabolites (Table 3), through pathway impact analysis, we were able to determine that the number of significantly dysregulated metabolic pathways was evenly distributed among treatments, ranging from two to three pathways. Even though the number of dysregulated metabolites varies, the number and identity of metabolic pathways impacted are similar. Among these pathways, lipid metabolism seems to be the main aspect affected, particularly glycerophospholipid and sphingolipid metabolism, although evidence of alteration of other pathways was also found, with lower impact. For TNBC cells, the sphingolipid metabolism impact score was lower than that for LA cells; for TNBC cells, glycerophospholipid metabolism was the most dysregulated. In contrast, sphingolipid metabolism was followed by glycerophospholipid metabolism as the most dysregulated pathway for LA cells. As stated by Zhao et al. (2013) and Nagarajan et al. (2021), modifications in lipid metabolism are one of the major components that enable cancer cells to adapt, which allows them to generate the necessary components for their development and tumorigenesis, such as increases in lipid uptake, lipogenesis, membrane synthesis, or signaling processes through phospholipids [50,51,52]. In another study, T. Burg et al. (2021) demonstrated that there is a relationship between epigenetic modifications and lipid metabolism defects in the spinal cord of a model of amyotrophic lateral sclerosis (ALS) in FUS mice according to transcriptomic and lipidomic analysis of treatment with a hydroxamic-derived selective HDAC6 inhibitor, ACY-738 [53]. The correlation of the previous studies with our results highlights the relevant role of the inhibition of HDACs in the metabolic modification of fatty acids in cancer cells, which, as is well known, are capable of reprogramming for their structural benefit and energy [54].

As mentioned above, HO-AAVPA was designed considering the VPA and hydroxyarylamide pharmacophore groups to optimize the HDAC inhibitory effects; later, it was also found that HO-AAVPA possessed some other VPA effects but with better results, such as anticonvulsant activity, antiproliferative effects, promotion of translocation of HMGB1 protein, production of ROS imbalance, and modification of the acetylation state of proteins, which have led to the idea that both compounds share a number of similar mechanisms of action but HO-AAVPA has lower toxicological and teratogenic profile effects [32,35,36,37]. The evidence of similarity between HO-AAVPA and VPA is further reinforced by our metabolomics analysis, which obtained identical impact scores for HO-AAVPA and VPA with TNBC cells and quite similar scores for LA cells for glycerophospholipid and sphingolipid metabolism (Table 4). Thus, we generated three sets of results for each cell line: (1) the metabolic pathways that were affected by both compounds in a meaningful way (high impact score); (2) metabolic pathways that were differentially affected; and (3) metabolic pathways with no impact (low-impact score), but whose presence might give us some information when compared to other studies.

Zhou et al. (2019) investigated the effect of VPA on the BC cell lines MDA-MB-231 and MCF-7 via a metabolomics approach. Among the multiple pathways on which VPA exerted its effect, they found sphingolipid metabolism within their MCF-7 dataset, although with a nonsignificant impact, contrary to what we found for the same pathway and both cell lines [44]. Interestingly, several pathways that were considered to have no impact (aminoacyl-tRNA biosynthesis, beta-alanine metabolism, pantothenate, and CoA biosynthesis, alanine, aspartate, glutamate, pyrimidine, and purine metabolisms) in our MCF-7 pathway analysis of HO-AAVPA-treated cells were found to be impactful in VPA-treated cells, which might indicate that HO-AAVPA and VPA indeed share mechanisms to exert their antiproliferative effect [44]. For MDA-MB-231, we found no similarities between our pathway analysis and the one performed by Zhou et al. (2019). In fact, these authors discovered that taurine and hypotaurine metabolism and beta-alanine metabolism are altered in both cell lines after VPA treatment. Additionally, alanine, aspartate, and glutamate metabolisms are mainly affected in MCF-7 cells and pyrimidine metabolism in MDA-MB-231 cells [44].

Estrada-Pérez et al. (2022) evaluated VPA in MCF-7 cells and reported the downregulation of members of important pathways, mainly the nonoxidative branch of the pentose phosphate pathway (PPP) and 2′deoxy-α-D-ribose-1-phosphate degradation, by administering the corresponding IC_50_ concentration of VPA [43,44]. Although none of these pathways were altered in our experiments, PPP was altered in the experiments performed by Zhou et al. (2019) in the same line using a concentration of 4 mM, halfway between our calculated IC_15_ and IC_50_ for that cell line (Table 1), so treating MCF-7 cells with VPA might first affect lipid metabolism, and as the VPA concentration increases, the pathways affected widen [44].

Gomes et al. (2020) also found that administering a moderate HDACi, resveratrol, modifies the lipid composition of MDA-MB-231 and MCF-7 cell lines [55]. Resveratrol decreases the biosynthesis of phosphatidylcholine (PC), phosphatidylinositol (PI), and lysophosphatidylcholine (LPC) in both cell lines, while phosphatidylethanolamine (PE) is also decreased in MDA-MB-231 cells and increased in MCF-7 cells [55]. Both HO-AAVPA and VPA also downregulated PC in both cell lines; PE was downregulated in all cases, except in MDA-MB-231 cells treated with VPA; LPC and lysophosphatidylethanolamine (LPE) were also downregulated, but such changes were detected only in MCF-7 cells treated with HO-AAVPA. Resveratrol is considered a pan inhibitor of HDAC [56,57], which has major activity on isoforms 1, 10, 4, and 9 of HDAC [56], all inhibited by VPA as well, except for HDAC10, so this might explain the similarities found between HDACi and VPA and HO-AAVPA [7].

The effect of VPA on different lipids has been reported in other contexts. Xu et al. (2019) evaluated the effect of VPA on lipids such as PE, PC, LPE, LPC, PI, sphingomyelin (SM), diacylglycerol (DG), and triacylglycerol (TG) and participants in sphingolipid metabolism, such as ceramides, from patients’ serum with abnormal liver function (ALF). Additionally, LPC, SM, and ceramides showed a significant decrease in concentration [58]. We found a higher number of dysregulated ceramides (upregulated) in MCF-7 cells than in MDA-MB-231 cells (downregulated) due to the effect of HO-AAVPA, while only one ceramide was found to be downregulated in MCF-7 VPA-treated cells. Xu et al. (2019) also evaluated the intracellular lipid content of L02 hepatic cells and observed an increase in lipid content due to VPA, which highlights the role of lipids in the effect of VPA and, in consequence, HO-AAVPA. They also noticed an increase in TG levels in L02 cells in a concentration-dependent manner, which we also observed as eight upregulated TGs in MCF-7 cells treated with HO-AAVPA.

Arachidonic acid (AA) was observed to have an upregulated tendency as a result of the treatment of MDA-MB-231 with VPA. The effect of VPA on AA has been studied before by Shimshoni et al. (2011) for bipolar disorder; in their work, they found that VPA inhibited the conversion of AA into acyl-CoA [58,59]. A similar event could occur in MDA-MB-231 cells exposed to VPA, provoking its accumulation seen as upregulation, a mechanism not found in HO-AAVPA-treated cells. Borin et al. (2017) focused on the AA pathway to collect information in their review of its implication on BC cell migration and invasion capacity; they noted that both invasion and metastasis decrease through the inhibition of the synthesis of 20-hydroxy-eicosatetraenoic acid (20-HETE) [60], so AA pathway dysregulation through VPA could be relevant to its antiproliferative effect in MDA-MB-231 cells treated with VPA but not in HO-AAVPA-treated cells.

As mentioned by Rosario et al. (2018) in their research on dysregulated pathways using transcripts from tumor and nontumor samples, the pentose glucuronate interconversion (PGI) pathway seems to be a commonly and significantly dysregulated pathway across multiple types of cancer, including BC [61], and this specific alteration may be how HO-AAVPA exerts its effect on MCF-7 cells. In the same study, the authors explored changes in the metabolism of BC cells versus normal cells and detected that glycerophospholipid and AA metabolism are dysregulated in BC cells, while sphingolipid and alpha-linolenic acid are not.

The differences observed between the studies mentioned above might be due to differences in the sample preparation methods, the compound concentration used, the exposure time to the compound, storage, data acquisition methods (including the acquisition platform), and other factors, as has been noted in other publications [62,63,64].

Although the scope of this work is limited to the putative annotation of metabolites, and definitive confirmation of metabolite identity requires direct comparison with reference compounds (metabolite identification confidence level 1, identity validation) [42], our experimental and analytical strategy used strict criteria to ensure high-quality annotations and conclusions, preventing false positives. This strategy provides us with a rigorous way to evaluate the possible identity of the features found. Results gathered through this study reveal interesting insights that required further confirmation by a targeted approach. Both the direction of dysregulated metabolites and the impact on pathways need to be addressed in further studies complemented with studies like proteomics.

## 3. Materials and Methods

### 3.1. Chemicals and Materials

Cell culture plastic material was purchased from TPP (Trasadingen, Switzerland), and fetal bovine serum (FBS) and trypsin-EDTA were acquired from Biowest (Riverside, MO, USA). LC-MS grade methanol was purchased from Honeywell Burdick and Jackson (Morristown, NJ, USA); 2-propanol and sodium hydroxide were purchased from Merck (Toluca, Mexico); chloroform, HEPES, urea, bicinchoninic acid, and sodium tartrate dihydrate were purchased from Sigma-Aldrich (Toluca, Mexico); sodium carbonate and sodium bicarbonate were purchased from Fermont (Mexico City, Mexico); and cupric sulfate pentahydrate was purchased from Golden Bell (Mexico City, Mexico). Ultrapure water was obtained from a Direct-Q 3 system (Millipore, Burlington, MA, USA). HO-AAVPA was synthesized as published by Prestegui-Martel et al. (2016) with a few modifications [37]. VPA, formic acid, ammonium acetate, and ammonium formate were acquired from Sigma-Aldrich (Toluca, México).

### 3.2. Cell Culture

MCF-7 and MDA-MB-231 cells were kindly donated by Dr. Gisela Ceballos Cancino (INMEGEN, Mexico City, Mexico). Both cell lines were grown in Dulbecco’s Modified Eagle’s Medium/High Modified (DMEM) without phenol red supplemented with 10% fetal bovine serum (FBS) at 37 °C and 5% of CO_2_ in a humidified atmosphere [43]. Cells were handled in a LabGard ES NU-540-400, Class II, Type A2 Laminar Flow (NUAIRE, Plymouth, MN, USA). The cells were detached using trypsin-EDTA upon reaching 75 ± 5% confluence, counted using a CytoSMART (Corning, Glendale, AZ, USA), and seeded in 60.1 cm^2^ tissue culture dishes until the required cell number was reached for the corresponding experiment.

### 3.3. Inhibitory Concentration Assays

Inhibitory concentration (IC) determinations were measured through MTT (3-(4,5-dimethylthiazol-2-yl)-2,5-diphenyltetrazolium bromide) assays in triplicate. Briefly, 1 × 10^4^ cells in 100 µL of media were seeded in each well of a 96-well tissue culture plate and incubated for 24 h. Afterward, the culture media was replaced with media with one of nine concentrations evaluated, from 140.625 µM to 36 mM VPA and 9.375 µM to 2.4 mM HO-AAVPA, in 150 µL of media, and the cells were incubated for 72 h. HO-AAVPA was first dissolved in DMSO up to a final concentration of 0.1% in culture media; 0.1% DMSO in culture media was used as a diluent control. After the incubation period, the medium was replaced with 100 µL of MTT solution in PBS (0.5 mg/mL), incubated for 4 h, and then replaced with DMSO. Absorbance was registered at 550 nm in a Multiskan Sky with Cuvette and Touch Screen (Thermo Scientific, Waltham, MA, USA) spectrophotometer with 15 s of agitation. IC_15_ was calculated with GraphPad Prism 8 (version 8.01 for Windows) using the equation for log (inhibitor) versus response variable slope (four parameters).

### 3.4. Cell Treatment and Metabolite Extraction

For metabolomics, assays with seven independent replicates were conducted. For each replicate, four 147.8 cm^2^ tissue culture dishes with 7 × 10^6^ viable cells in 13 mL culture media were prepared for the control, diluent control (DMSO 0.1%), VPA, and HO-AAVPA samples (IC_15_ treated to normalize effects for both compounds and cell lines) [44,63]. Before any treatment, the cells were allowed to grow for 24 h, and then the medium was replaced with the corresponding treatment and incubated for 48 h. VPA and HO-AAVPA cells were treated with 3.48 and 1.64 mM VPA and 175.6 and 313.8 µM HO-AAVPA for MDA-MB-231 and MCF-7, respectively. An additional extraction control was prepared in the same way as the diluent control, except that no cells were seeded in it.

Metabolite extraction was based on the Bligh–Dyer method for polar and nonpolar compounds, reported elsewhere [65], with modifications proposed by Agilent Technologies (Santa Clara, CA, USA) application note 5991-3528EN and a randomized extraction order. Briefly, cells were maintained in wet ice; the media was discarded, and then the cells were washed three times with 10 mL of 0.9% NaCl. Liquid nitrogen was applied to the monolayer to stop cell metabolism. Then, 2 mL of methanol was added, and the cells were scraped with a cell scraper. Cells were recovered in a 2 mL plastic vial and maintained in dry ice until all cells were harvested; then, the samples were transferred to −80 °C storage for further processing. Once all replicates were obtained, 50 µL of a solution of acetaminophen (1200 ppm) and carbamazepine (1200 ppm) was added to each sample, mixed, sonicated with a Vibra-Cell VC 130 Ultrasonic Processor (Sonics and Materials, Newtown, CT, USA) by applying pulses with a frequency of 40 kHz with an on and off cycle of 5 and 1 s, respectively, five times, and then the samples were split in two. Sequentially, 250 µL of chloroform, 350 of water, and 250 µL of chloroform were added and mixed for 10 s in a vortex after each addition. Separation of phases was achieved by centrifugation at 5000 rpm at 4 °C for 30 min. The aqueous and organic phases of samples that were previously split were merged in plastic vials and stored at −80 °C (including the protein disc). The organic phase and protein disc (for protein quantitation) were dried in an orbital incubator (INO-650 M, Prendo, Puebla, Mexico) at 30 °C, and the aqueous phase was dried in a Vacufuge plus (Eppendorf, Hamburg, Germany). Once dried, the samples were stored at −80 °C for further analysis.

### 3.5. Protein Quantification

Protein was quantitated by the bicinchoninic acid method [66]. Briefly, a nine-level calibration curve (0, 25, 125, 250, 500, 750, 1000, 1500, and 2000 µg/mL BSA) was prepared by serial dilution (diluent solution: 8 M urea and 20 mM HEPES, pH 8) [67]. Protein discs from MDA-MB-231 and MCF-7 cells were diluted in 1.0 and 1.6 mL of diluent solution, respectively, and mixed for 2 min with a vortexer. Fifty microliters of protein solution from MCF-7 cells were further diluted in 150 µL of diluent. Then, 25 µL of each level of calibration curve or sample (in duplicate) was added to a 96-well flat bottom plate (Corning) and mixed with 200 µL of working solution. The plates were incubated in an orbital incubator with constant agitation at 37 °C for 30 min in darkness. Absorbance values were registered at 562 nm with 10 s of agitation in a Multiskan Sky with Cuvette and Touch Screen (Thermo Scientific) spectrophotometer [68]. The working solution was prepared by mixing 50 parts of reagent A solution (0.1 g bicinchoninic acid, 2 g sodium carbonate, 0.16 g sodium tartrate dihydrate, 0.4 g sodium hydroxide and 0.95 g sodium bicarbonate in 100 mL of water) and 1 part of reagent B solution (0.4 g cupric sulfate pentahydrate in 10 mL of water). The protein concentration of each sample was obtained by substituting the corresponding absorbance values in the equation of the straight line.

### 3.6. LC-MS Data Acquisition

Data acquisition for each cell line was performed separately using an UHPLC 1290 Infinity II: 1290 Flexible Pump (G7104A) and 1290 vial sampler with integrated column compartment (G7129B), coupled with a Q-TOF (G6545A) with Dual AJS ESI as ionization source (G1959A), all from Agilent Technologies. A total of four acquisition conditions were used: HILIC-ESI(+)-MS; HILIC-ESI-(−)-MS (hydrophilic interaction liquid chromatography for polar metabolites); RPLC-ESI(+)-MS; and RPLC-ESI-(−)-MS (reversed-phase liquid chromatography for nonpolar metabolites). The sample injection order was as follows: the LC–MS system was first equilibrated by injecting 10 µL of the blank sample until no chromatographic variation was observed (blank injection); then, blank and blank + standards were injected twice. Afterward, an extraction blank was injected in triplicate, and later, to equilibrate the LC–MS system to biological samples, quality control samples (QC) were injected until no chromatographic variation was observed. QC samples were injected before and after every five biological sample injections, and the sequence order of the biological samples was randomly assigned [69,70].

Polar compounds were separated using a Poroshell 120 HILIC-Z (2.1 × 150 mm, 2.7 µm) column with a Poroshell 120 HILIC-Z (2.1 × 5 mm, 2.7 µm) guard column (Agilent Technologies, Santa Clara, CA, USA) through a nonlinear gradient for both positive and negative acquisition (application note: 5994-1492EN, Agilent Technologies, for positive and negative LC and MS acquisition). MDA-MB-231 cells were resuspended in 120 µL of ACN/MeOH/H_2_O 70:20:10 (*v*/*v*/*v*) (HILIC diluent), and MCF-7 cells were first diluted in 150 µL of HILIC diluent and mixed. Then, 12.5 µL of this mixture was further diluted with 47.5 µL of HILIC diluent to obtain similar signal intensities between MDA-MB-231 and MCF-7 metabolites. Positive acquisition used ammonium formate (AmF) 10 mM and formic acid (FA) 0.125% in water (solvent A) and AmF 10 mM and FA 0.125% in ACN/H_2_O 90:10 (*v*/*v*) (solvent B) as follows: 0–3 min of 98% B; 70% B at 11 min; 60% B at 14 min; and 10% B from 18 to 20 min with 4 min of post time. The column temperature was maintained at 25.0 ± 0.5 °C. Negative acquisition used ammonium acetate (AmAc) 10 mM in water (solvent A) and AmAc 10 mM in ACN/H_2_O 85:15 (*v*/*v*) (solvent B) as follows: 0–2 min of 92% B; 84% B from 5.5 to 8.5 min; 82% B from 9 to 14 min; 78% B at 17 min; 65% B at 21 min; 40%% B from 23 to 25 min; and 5% B from 27 to 29 min with 4 min of post time. The column temperature was maintained at 25.0 ± 0.5 °C and 40.0 ± 0.5 °C for positive and negative acquisition, respectively. The flow rate was 0.25 mL/min with 10 µL of volume injected and 15 s of needle washing with methanol for both cases.

The spectrometric conditions for polar compounds were as follows: drying gas temperature of 225 and 225 °C; drying gas flow of 8 and 13 L/min; sheath gas temperature of 225 and 350 °C; sheath gas flow of 10 and 12 L/min; nebulizer pressure of 40 and 35 psig; capillary voltage of 3000 and 3500 V; nozzle voltage of 0 and 0 V; fragmentor of 125 and 125 V; skimmer of 65 and 45 V; and octupole RFF of 450 and 750 V for positive and negative acquisition, respectively. The scan rate was 3 spectra/s and 50–1700 *m*/*z* for mass range; correction was performed with 121.05087300 and 922.00979800 *m*/*z* and 68.99575800 and 1033.98810900 *m*/*z* as mass references for positive and negative acquisition, respectively.

Nonpolar compounds were separated using a Zorbax Eclipse Plus C18 (2.1 × 150 mm, 1.8 µm) column with a Zorbax Eclipse Plus C18 (2.1 × 5 mm, 1.8 µm) guard column (Agilent Technologies) through a nonlinear gradient for both positive and negative acquisition [65]. MDA-MB-231 cells were resuspended in 120 µL of IPA/ACN 90:10 (*v*/*v*) (RPLC diluent); MCF-7 cells were first diluted in 150 µL of RPLC diluent and mixed. Then, 12.5 µL of this mixture was further diluted with 47.5 µL of RPLC diluent to obtain similar signal intensities between MDA-MB-231 and MCF-7 metabolites. Positive acquisition used AmF 10 mM and FA 0.1% in ACN/H_2_O 60:40 (*v*/*v*) (solvent A) and AmF 10 mM and FA 0.1% in IPA/ACN 90:10 (*v*/*v*) (solvent B) as follows: 32% B at 0 min; 40% B from 2 to 3 min; 45% B at 8 min; 50% B at 10 min; 60% B at 16 min; 70% B at 22 min; and 90% B from 28 to 36 min with a 3 min post time. Negative acquisition used 10 mM AmAc in 60:40 (*v*/*v*) ACN/H2O (solvent A) and 10 mM AmAc in 90:10 (*v*/*v*) IPA/ACN (solvent B) and the same gradient configuration as in positive acquisition. The column temperature was maintained at 60.0 ± 0.5 °C; the injection volume was 2 µL; the flow rate was 0.3 mL/min, and the column was washed with IPA/ACN 90:10 (*v*/*v*) for both acquisitions.

The spectrometric conditions for nonpolar compounds were as follows (application note: 5991-9280EN, Agilent Technologies): drying gas temperature of 200 °C; drying gas flow of 13 L/min; sheath gas temperature of 350 °C; sheath gas flow of 11 L/min; nebulizer pressure of 35 psig; capillary voltage of 3500 V; nozzle voltage of 1000 V; fragmentor of 175 V; skimmer of 65 V; voltage octupole RFF of 750 V; scan rate of 3 spectra/s and 50–1700 *m*/*z* for mass range for positive and negative acquisition, respectively; correction was performed with 121.05087300 and 922.00979800 *m*/*z* and 68.99575800 and 1033.98810900 *m*/*z* as mass references for positive and negative acquisition, respectively.

### 3.7. LC-MS Data Processing

Optimization of the feature extraction parameters for the Molecular Feature Extraction (MFE) algorithm, the retention time drift tolerance, ionic species included, and compound threshold were determined with MassHunter Workstation Software Qualitative Analysis (version B.07.00, build 7.7.7024.29, SP2, Agilent Technologies, Santa Clara, CA, USA) [71]. These parameters were optimized for each condition acquisition and for each cell line separately through inspection of the corresponding initial QC. Once the extraction parameters were optimized, all blank samples were first batch-analyzed separately to generate an exclusion feature list (a list with features present in the blank, except added standards and mass references, that need to be excluded from the analysis of samples) and then all QCs and biological samples were batch-analyzed, excluding features present in the exclusion list. Batch alignment and extraction were performed with MassHunter Profinder (version B.08.00, SP3, Agilent Technologies, Santa Clara, CA, USA) using the Batch Recursive Feature Extraction for small molecule/peptide algorithm using common organic molecules (no halogens) as an isotopic model, allowing for a maximum of two charges and only compounds with two or more ions to avoid false-positives. To further reduce false positives, only compounds present in four and six files (for the MFE and Find by Ion or FBIon algorithms, respectively) in at least one sample group (pseudo-replicates) and with an MFE score ≥ 70.0 were retained for further analysis and exported as a profinder archive (.pfa with median mass and retention time). For features present in blanks, only those present in one file and 100% of files in at least one sample group were retained in the MFE and FBIon algorithms, respectively. The results were inspected, particularly the correct integration of added standards. Before batch alignment, a time alignment was performed for all samples (blanks and QCs included) versus the initial QC to further correct retention time drift as follows: features with counts higher than 1000; maximum time shift of 0.5 min + 0.5%; and polynomial interpolation as a fitting model to reduce variation in the retention time.

Chemometric comparisons between all treatments for a given cell line were performed in Mass Profiler Professional (MPP) (version 14.9.1, Agilent Technologies, Santa Clara, CA, USA). First, normalization by internal standard (carbamazepine for HILIC-ESI(+) and acetaminophen for the rest of the conditions) and external scalar (protein concentration) were performed to correct for variation due to the extraction process and quantity of biological material, respectively. Then, principal component analysis (PCA) was performed on all samples to be grouped. Afterward, entities (features are called entities in MPP) were filtered by variability, only keeping those with standard deviation ≤ 1.0 for all conditions (treatments). The Shapiro–Wilk (*p*-value of 0.05) normality test was carried out, and two entity lists were created, one for entities with normal distribution and the other for entities with nonnormal distribution. Entities within the first list were analyzed with one-way ANOVA, Tukey’s HSD post hoc test, asymptotic computation of *p*-value, and multiple testing correction Benjamini Hochberg FDR. Entities within the second list were analyzed with the Kruskal–Wallis test, asymptotic computation of *p*-value, and multiple testing correction Benjamini Hochberg FDR. Pairwise contrast was performed by contrasting the treatments with their respective control, culture with medium for VPA, and vehicle control (DMSO) for HO-AAVPA. Only entities with a *p*-value ≤ 0.01 and a fold change value ≥ 2.0 were kept for both lists, excluding missing values for calculation. Entities were used to create an inclusion list, namely, a list of ions that are fragmented during LC-MS/MS data acquisition.

### 3.8. LC-MS/MS Data Acquisition and Metabolite Annotation

The first LC-MS/MS data acquisition was performed with the same chromatographic and spectrometric conditions previously described but with the following modifications: auto MS/MS mode with a list of preferred ions (features found to be relevant in the statistical analysis) by cell line; fragmentation under three different collision energies (10, 20 and 40 V); a number of charges = 1 (z = 1); isolation window width of ~4 amu; and retention time delta of 0.3 min. The scan rate for MS/MS was kept at 3 spectra/s in a range of 50–1700 *m*/*z*. This second data acquisition was performed by injecting 3 µL of the corresponding cell line quality control sample [43,71].

Scan spectra and extracted compound chromatograms from fragmented compounds were recovered through MassHunter Qualitative Analysis’s Find by Auto MS/MS algorithm, as well as product ion spectra by collision energy. Product ion spectra abundances were transformed from counts to percentage values and then searched against the HMDB 5.0 [72] using the LC-MS/MS Search tool (https://hmdb.ca/spectra/ms_ms/search, accessed from 1 April 2022 to 30 June 2022) as follows: low, medium, and high energy (10, 25 and 40 V, respectively); parent ion mass and mass/charge tolerance of 10 ppm; and enabling the inclusion of predicted spectra. The results with Fit, RFit, and Purity values ≥ 0.80 were kept, and the result with the highest value was assigned to annotate the corresponding compound as a putative metabolite. The annotation information for each compound was updated using the IDBrowser tool from MPP (version 14.9.1, Agilent Technologies, Santa Clara, CA, USA).

### 3.9. Effect on Metabolic Pathways

To determine the corresponding compound’s impact on metabolic pathways, we used the Metaboanalyst 5.0 platform with the Pathway Analysis module (https://dev.metaboanalyst.ca/MetaboAnalyst/upload/PathUploadView.xhtml, accessed on 1 July 2022 to 31 July 2022) [49]. HMDB accession numbers were used for the search (with the HMDB ID as the input type); scatter plots were used as the visualization method; the hypergeometric test was applied as the enrichment method; relative betweenness centrality was applied as topology analysis, and Homo sapiens (KEGG) was used as the pathway library [73,74,75,76,77].

## 4. Conclusions

In the present study, we tentatively identified the metabolic pathways modulated by two structurally similar HDACi, HO-AAVPA and VPA, in BC cells. The results highlight their main effect on metabolic pathways related to lipid metabolism in MCF-7 and MDA-MB-231 cell lines, supporting the hypothesis of similar mechanisms of action, but other deregulated metabolites found suggest the presence of different biological targets. However, a proteomic approach is required to support this claim. Furthermore, the observed differences in the metabolic pathways altered by each compound may be key to understanding how HO-AAVPA enhances VPA potency. Future studies are required to confirm the identity of the putative metabolites reported in this work. Despite these limitations, these findings provide a solid basis for understanding the mechanisms of metabolic regulation exerted by two HDACis and offer valuable insights for developing therapeutic strategies in breast cancer.

## Figures and Tables

**Figure 1 ijms-24-14543-f001:**
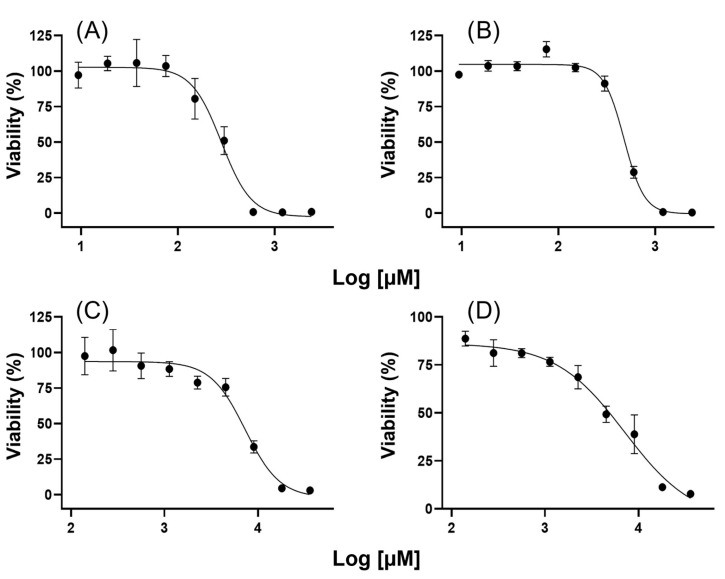
The antiproliferative effects of different concentrations of HO-AAVPA and VPA on cell viability. (**A**) Effect of HO-AAVPA on MDA-MB-231 cells; (**B**) Effect of HO-AAVPA on MCF-7 cells; (**C**) Effect of VPA on MDA-MB-231 cells; (**D**) Effect of VPA on MCF-7 cells.

**Figure 2 ijms-24-14543-f002:**
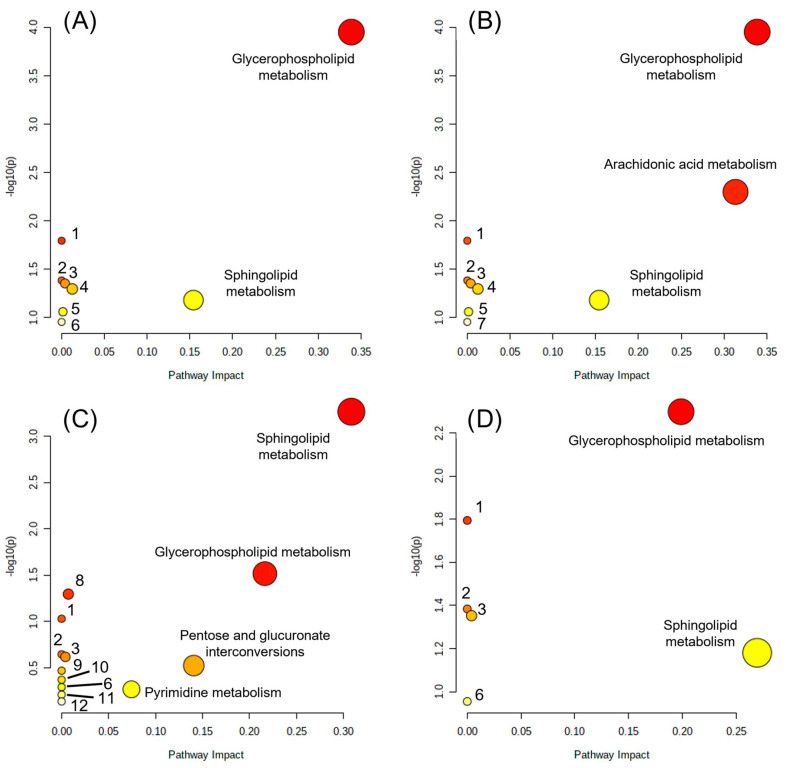
Pathway impact analysis. (**A**) HO-AAVPA on MDA-MB-231; (**B**) VPA on MDA-MB-231; (**C**) HO-AAVPA on MCF-7; (**D**) VPA on MCF-7. Linoleic acid metabolism (1), alpha-linolenic acid metabolism (2), glycosylphosphatidylinositol (GPI)-anchor biosynthesis (3), glycerolipid metabolism (4), phosphatidylinositol signaling system (5), arachidonic acid metabolism (6), biosynthesis of unsaturated fatty acids (7), pantothenate and CoA biosynthesis (8), beta-alanine metabolism (9), alanine, aspartate, and glutamate metabolism (10), aminoacyl-tRNA biosynthesis (11), purine metabolism (12). The circle size correlates with the corresponding x-axis value, and the color (from yellow to red) correlates with the corresponding y-axis value.

**Table 1 ijms-24-14543-t001:** IC values of HO-AAVPA and VPA in triple-negative breast cancer and luminal breast cancer cells.

	MCF-7		MDA-MB-231	
	IC_50_	IC_15_	IC_50_	IC_15_
HO-AAVPA	476.1 µM	313.8 µM	291.8 µM	175.6 µM
VPA	7.11 mM	1.64 mM	7.29 mM	3.48 mM

**Table 2 ijms-24-14543-t002:** Putative metabolites dysregulated in MDA-MB-231 cells.

Metabolite	Change in Regulation			
	HO-AAVPA	Fold Change	VPA	Fold Change
Sphinganine	Up *	2.583	Up *	2.295
4,5-Dihydro-1-benzoxepin-3(2H)-one	Down *	−16	Up *	16
N-trans-Feruloyloctopamine	Up *	16	Down *	−16
PC(18:4(6Z,9Z,12Z,15Z)/14:0)	ND	-	Up *	16
PE(15:0/14:0)	Down *	−16	Up *	11.739
PE(14:0/15:0)	Down *	−16	Up *	14.129
PS(18:0/18:0)	Down *	−4.61	Down	−1.06
PC(14:0/14:0)	Down *	−148.631	Up *	16
TG(15:0/22:6(4Z,7Z,10Z,13Z,16Z,19Z)/22:6(4Z,7Z,10Z,13Z,16Z,19Z))	Down *	−16	Up *	16
TG(15:0/20:4(5Z,8Z,11Z,14Z)/22:6(4Z,7Z,10Z,13Z,16Z,19Z))	Down *	−34.84	Up *	16
TG(14:0/22:4(7Z,10Z,13Z,16Z)/14:0)	Up *	2.138	Up	1.539
Cer(d18:0/26:1(17Z))	Down *	−3.127	Down	−1.312
Cer(d18:0/26:0)	Down *	−2.734	Down	−1.219
TG(20:2n6/18:0/20:2n6)	Up *	2.145	Up	1.362
Aspidospermatine	Up *	26.008	ND	-
Arachidonic acid	Down	−1.876	Up *	4.491
PA(16:0/16:0)	Down *	−16	Up *	16
PE(14:0/22:5(4Z,7Z,10Z,13Z,16Z))	Down *	−3.773	Up	1.697
PE(20:0/14:0)	Down *	−16	Up *	16

*: significantly dysregulated FDR adjusted *p* ≤ 0.01; up: upregulated; down: downregulated; ND: not detected.

**Table 3 ijms-24-14543-t003:** Putative metabolites dysregulated in MCF-7 cells.

Metabolite	Change in Regulation			
	HO-AAVPA	Fold Change	VPA	Fold Change
LysoPC(14:0)	Up *	3.569	Up	1.012
LysoPC(16:0)	Up *	3.198	Up	1.116
PC(14:0/14:0)	Up	1.374	Up	1.091
PC(18:2(9Z,12Z)/18:4(6Z,9Z,12Z,15Z))	Up *	3.073	Down *	−2.196
CL(18:1(9Z)/16:0/18:1(9Z)/18:0)	Up *	3.145	Down *	−2.198
PE(22:6(4Z,7Z,10Z,13Z,16Z,19Z)/16:1(9Z))	Up *	2.428	Down *	−2.189
PC(18:2(9Z,12Z)/18:3(6Z,9Z,12Z))	Up *	2.845	Down *	−2.167
PC(18:3(6Z,9Z,12Z)/16:0)	Up	1.696	Down *	−2.201
PE(14:0/22:5(4Z,7Z,10Z,13Z,16Z))	Up	1.706	Down *	−2.129
PE(22:5(7Z,10Z,13Z,16Z,19Z)/14:0)	Down *	−2.883	Up	1.098
PE(20:3(8Z,11Z,14Z)/14:0)	Down *	−2.820	Down	−1.220
Cohibin D	Down *	−2.297	Up	1.054
PC(16:0/15:0)	Down *	−2.311	Down	−1.223
PC(14:1(9Z)/15:0)	Down *	−1.964	Down	−1.078
CL(18:1(9Z)/18:0/18:2(9Z,12Z)/18:0)	Down *	−2.034	Down	−1.118
PE(20:2(11Z,14Z)/14:0)	Down *	−2.742	Down *	−2.310
PC(18:3(6Z,9Z,12Z)/18:0)	Down *	−2.363	Down *	−2.261
PC(18:3(6Z,9Z,12Z)/15:0)	Down *	−2.335	Down *	−2.078
PC(20:5(5Z,8Z,11Z,14Z,17Z)/15:0)	Down *	−2.358	Up *	2.086
PC(15:0/18:3(9Z,12Z,15Z))	Down	−1.501	Up *	2.053
PC(18:1(9Z)/18:1(9Z))	Down *	−2.322	Down *	−2.243
PC(16:1(9Z)/15:0)	Down	−1.831	Down *	−2.049
Ceramide (d18:1/16:0)	Up *	5.668	Down *	−2.323
PC(18:2(9Z,12Z)/15:0)	Down *	−2.479	Down *	−2.321
PC(18:2(9Z,12Z)/20:1(11Z))	Down *	−2.170	Up	1.063
PC(22:5(7Z,10Z,13Z,16Z,19Z)/20:0)	Down *	−2.915	Up	1.005
PE(24:1(15Z)/18:4(6Z,9Z,12Z,15Z))	Down *	−2.496	Down	−1.149
PC(22:5(7Z,10Z,13Z,16Z,19Z)/22:1(13Z))	Down *	−2.299	Up	1.005
PC(18:2(9Z,12Z)/20:0)	Down *	−2.582	Up	1.011
SM(d18:0/26:1(17Z))	Down *	−2.185	Up	1.113
Cer(d18:1/24:1(15Z))	Up *	2.078	Down	−1.322
Cer(d18:1/24:0)	Up *	3.452	Down	−1.427
Glucosylceramide (d18:1/26:0)	Down *	−4.280	Up	1.012
PE(18:0/24:1(15Z))	Down *	−3.980	Up	1.080
Cer(d18:1/26:0)	Down *	−2.594	Up	1.265
TG(18:4(6Z,9Z,12Z,15Z)/16:0/18:4(6Z,9Z,12Z,15Z))	Up *	16	ND	-
TG(14:1(9Z)/16:0/14:1(9Z))	Up *	4.823	Down	−1.645
CL(18:1(9Z)/16:1(9Z)/18:1(9Z)/16:1(9Z))	Down *	−3.829	Down	−1.068
TG(14:0/14:0/20:4(5Z,8Z,11Z,14Z))	Up *	5.439	Down	−2.203
TG(14:1(9Z)/15:0/16:1(9Z))	Up *	2.788	Down	−1.519
TG(16:1(9Z)/14:0/18:3(9Z,12Z,15Z))	Up *	10.634	Down	−1.531
TG(20:3n6/14:0/18:3(9Z,12Z,15Z))	Up *	7.207	Down	−1.182
Ubisemiquinone	Down *	−3.178	Down	−1.199
TG(16:1(9Z)/14:1(9Z)/18:1(9Z))	Up *	3.491	Down	−1.421
SM(d18:1/26:0)	Up *	8.390	Down	−1.036
TG(14:0/22:4(7Z,10Z,13Z,16Z)/14:0)	Up *	5.237	Down	−1.340
LysoPE(0:0/18:0)	Up *	3.069	Up	1.136
LysoPE(0:0/16:0)	Up *	2.538	Up	1.378
LysoPE(18:1(9Z)/0:0)	Up *	2.122	Up	1.092
LysoPE(18:0/0:0)	Up *	2.813	Up	1.281
PS(18:1(9Z)/16:0)	Up *	3.219	Up	1.086
Cer(d18:1/14:0)	Up *	9.224	Down	−1.176
PE(18:1(9Z)/14:0)	Down *	−2.021	Up	1.031
PE(14:1(9Z)/20:1(11Z))	Down *	−2.732	Down	−1.222
PE(14:1(9Z)/22:2(13Z,16Z))	Down *	−2.439	Down	−1.132
Cer(d18:0/14:0)	Up *	16.862	Down	−1.185
PE(22:2(13Z,16Z)/14:1(9Z))	Down *	−3.042	Down	−1.112
PE(16:1(9Z)/24:1(15Z))	Down *	−2.701	Down	−1.008
1,1’-(1,4-Dihydro-4-nonyl-3,5-pyridinediyl)bis [1-decanone]	Up *	2.518	Down	−1.265
PE(14:0/22:2(13Z,16Z))	Down *	−2.718	Down	−1.266
PE(24:0/14:0)	Down *	−3.026	Down	−1.131
Cer(d18:0/16:0)	Up *	4.630	Down	−1.096
PE-NMe(18:1(9Z)/18:1(9Z))	Down *	−2.462	Down	−1.224
‘PE(22:2(13Z,16Z)/16:1(9Z))’	Down *	−2.020	Up	1.204
PE(18:1(11Z)/24:1(15Z))	Down *	−2.565	Down	−1.001
2-O-(4,7,10,13,16,19-Docosahexaenoyl)-1-O-hexadecylglycero-3-phosphocholine	Down *	−2.393	Down	−1.235
PE(16:0/22:2(13Z,16Z))	Down *	−2.691	Down	−1.069
1-[1,4-Dihydro-4-nonyl-5-(1-oxodecyl)-3-pyridinyl]-1-dodecanone	Up *	3.533	Down	−1.016
PE(24:1(15Z)/14:0)	Down *	−2.070	Down	−1.055
PE(24:1(15Z)/16:1(9Z))	Down *	−2.355	Down	−1.321
1,1’-(1,4-Dihydro-4-nonyl-3,5-pyridinediyl)bis[1-dodecanone]	Up *	3.586	Up	1.236
Campesteryl linoleate	Up *	2.700	Down	−1.092
PE-NMe2(16:0/18:1(9Z))	Up *	2.601	Up	1.043
PE(24:1(15Z)/18:1(9Z))	Down *	−3.303	Down	−1.332
Cer(d18:0/24:1(15Z))	Up *	3.664	Down	−1.398
PE(24:1(15Z)/18:0)	Down *	−3.529	Down	−1.073
CL(18:2(9Z,12Z)/18:0/18:2(9Z,12Z)/16:1(9Z))	Down *	−2.271	Down	−1.359
Aprobarbital	Down *	−2.059	Down	−1.104
Dihydroandrosterone	Up *	2.259	Down	−1.115
Kanzonol O	Down *	−7.933	Down	−1.309
LysoPC(16:1(9Z))	Up *	4.319	Down	−1.171
1-Nitroheptane	Up *	2.624	Down *	−2.132
Styrene	Down *	−2.178	Down	−1.128
(S)-Homostachydrine	Up *	2.317	Down	−1.242
Buprenorphine	Up *	5.704	Down	−1.141
7-Methylguanosine	Up *	2.178	Up	1.147
L-Hexanoylcarnitine	Up *	15.273	Down	−1.139
cis-4-Decenedioic acid	Up *	16	ND	-
5-Aminoimidazole-4-carboxamide	Down *	−9.820	Down	−1.832
5-Butyltetrahydro-2-oxo-3-furancarboxylic acid	Down *	−2.074	Down	−1.133
4-Hydroxy-2-butenoic acid gamma-lactone	Down *	−2.895	Down	−1.077
Leucyl-Proline	Down *	−2.079	Down	−1.134
Pyro-L-glutaminyl-L-glutamine	Up *	16	ND	-
3’-O-Methyladenosine	Up *	23.771	Up	1.006
N-Ethylglycine	Down *	−2.040	Down	−1.029
N-Acetylglutamine	Down *	−6.166	Up	1.094
3-Acetamidobutanal	Up *	2.356	Up	1.010
Phenol sulphate	Down *	−2.135	Up	1.027
2-Methyl-3-ketovaleric acid	Down *	−2.801	Up	1.064
Palmitoyl glucuronide	Down *	−3.696	Down	−1.604
Pyrogallol-2-O-sulphate	Down *	−4.339	Up	1.237
Valdiate	Down *	−2.344	Down	−1.010
2,6-Di-tert-butyl-1,4-benzenediol	Down *	−2.020	Up	1.108
1-Hydroxy-2-pentanone	Down *	−2.406	Down	−1.179
Uracil	Down *	−2.504	Down	−1.095
3’-Hydroxy-3,4,5,4’-tetramethoxystilbene	Down *	−9.469	Down	−1.040
Acetoxyacetone	Down *	−2.190	Down	−1.304
Deoxyfructosazine	Down *	−11.270	Down	−1.260
Portulacaxanthin II	Down *	−10.825	Down	−1.309
Leucyl-Leucine	Down *	−3.482	Up	1.105
Homocysteinesulfinic acid	Down *	−2.240	Down	−1.147
N-Acetyl-L-methionine	Down *	−2.109	Up	1.257
Pantothenic acid	Up *	6.745	Down	−1.004
Phenylalanyl-Glycine	Down *	−6.639	Down	−1.099
Sakacin P	Down *	−5.802	Up	1.089
(±)-2,2’-Iminobispropanoic acid	Down *	−4.959	Up	1.089
Gamma-glutamyl-Phenylalanine	Down *	−3.403	Up	1.031
Racemethionine	Up *	6.745	Up	1.073
L-Aspartate-semialdehyde	Down *	−6.639	Up *	2.012
Phenylalanylglutamine	Down *	−5.802	Up *	2.017
L-N-(3-Carboxypropyl)glutamine	Down *	−4.959	Down	−1.067
Valyl-Alanine	Down *	−3.403	Down	−1.069
Alanyl-Glycine	Down *	−4.922	Down	−1.043
Glycyl-Hydroxyproline	Down *	−4.045	Up	1.269
Ethyl nitrite	Down *	−13.453	Down	−1.076
Methoxyacetic acid	Down *	−2.156	Up	1.272
L-Asparagine	Up *	16	ND	-
Acetophenazine	Down *	−2.405	Down *	−2.083

*: significantly dysregulated FDR adjusted *p* ≤ 0.01; up: upregulated; down: downregulated; ND: not detected.

**Table 4 ijms-24-14543-t004:** Pathway impact analysis of HO-AAVPA and VPA on TNB and LA cells.

Metabolic Pathway	MDA-MB-231		MCF-7	
	HO-AAVPA	VPA	HO-AAVPA	VPA
Glycerophospholipid metabolism	0.34	0.34	0.22	0.20
Sphingolipid metabolism	0.15	0.15	0.31	0.27
Arachidonic acid metabolism	0.00	0.31	0.00	0.00
Linoleic acid metabolism	0.00	0.00	0.00	0.00
alpha-linolenic acid metabolism	0.00	0.00	0.00	0.00
Glycosylphosphatidylinositol (GPI)-anchor biosynthesis	0.00	0.00	0.00	0.00
Glycerolipid metabolism	0.01	0.01	NA	NA
Phosphatidylinositol signaling system	0.00	0.00	NA	NA
Biosynthesis of unsaturated fatty acids	NA	0.00	NA	NA
Pantothenate and CoA biosynthesis	NA	NA	0.01	NA
Pentose and glucuronate interconversions	NA	NA	0.14	NA
Beta-alanine metabolism	NA	NA	0.00	NA
Alanine, aspartate, and glutamate metabolism	NA	NA	0.00	NA
Pyrimidine metabolism	NA	NA	0.07	NA
Aminoacyl-tRNA biosynthesis	NA	NA	0.00	NA
Purine metabolism	NA	NA	0.00	NA

NA: Not applicable.

## Data Availability

The data presented in this study are openly available in Zenodo at https://doi.org/10.5281/zenodo.7991779 since 31 July 2023.

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
