# Peer review of "Untargeted LC-MS/MS Metabolomics Study of HO-AAVPA and VPA on Breast Cancer Cell Lines"

_ijms, 2023, doi:10.3390/ijms241914543_

Round 1
Reviewer 1 Report (New Reviewer)
Major Issues:
Technique Validation and Clinical Relevance:
One major concern with the study is the reliance on only one technique, LC‒MS, to identify the metabolites. To establish the clinical relevance of these findings, it is essential to validate the results using a different technique or patient-derived breast cancer cells. While the authors mentioned the use of strict criteria to ensure high-quality annotations, this does not negate the need for confirmation with orthologous methods. I recommend performing additional experiments to address this limitation.
Confusing Results and Discrepancies:
The results presented in the manuscript are somewhat perplexing. For instance, the direction of fold change for the two drugs in the MDA-MB-231 cell lines is largely opposite. Furthermore, there is a substantial difference in the number of reported metabolites for MDA-MB-231 and MCF7 (as shown in Table 2 & 3), and this issue needs a more comprehensive explanation to help readers understand the findings better.
Inconsistencies in Impact Scores:
I noticed a discrepancy in the impact scores for HO-AAVPA and VPA in MDA-MB-231 cells. Despite obtaining identical impact scores for some aspects in the metabolomics analysis, the direction of fold change for most identified metabolites in MDA-MB-231 differs between the two drugs. This requires further clarification and analysis to resolve the inconsistencies.
Lack of Validation Methods and Clinical Relevance:
The study would greatly benefit from the inclusion of validation methods using alternative techniques and patient-derived materials. This would help establish the clinical significance of the findings and improve the robustness of the conclusions.
Minor Issues:
Introduction Simplification:
I suggest simplifying the introduction by deleting lines 31-36 and starting with "Breast cancer (BC) currently has the highest incidence (considering 37 males and females) and..."
Improved Clarity of Methods:
The description of the untargeted metabolomic study (Line 103-105) requires better clarity. Please rephrase this section to provide a more explicit explanation of the aims of the study, the "features" being referred to, and the utilization of fragmentation information for annotating corresponding putative metabolites.
In conclusion, while the topic of this manuscript is interesting and relevant, it is important to address the major issues concerning technique validation, result discrepancies, and the need for validation methods for clinical relevance. Moreover, the minor issues should be corrected to improve the overall clarity of the manuscript.
While the topic of this research is intriguing and relevant, I must bring attention to a significant concern regarding the use of the English language throughout the manuscript. There were instances where I found it challenging to comprehend the intended message (e.g., line 103-105, line 91-98, etc.), which hindered the overall readability and clarity of this work.
To improve the manuscript's quality and effectively convey the findings, I strongly recommend focusing on using simple and concise English language. This approach will help ensure that readers can easily understand and grasp the content without unnecessary complexities.
Author Response
Referee 1
Point 1. One major concern with the study is the reliance on only one technique, LC‒MS, to identify the metabolites. To establish the clinical relevance of these findings, it is essential to validate the results using a different technique or patient-derived breast cancer cells. While the authors mentioned the use of strict criteria to ensure high-quality annotations, this does not negate the need for confirmation with orthologous methods. I recommend performing additional experiments to address this limitation.
Response: Thank you for your suggestion. LC-MS is considered the predominant technique to study metabolomics due to its high metabolome coverage, resolving power, sensitivity and capacity to give orthogonal information like MS/MS spectra (fragmentation information, (https://doi.org/10.3390/metabo12040357), so it is considered as a powerful tool on its own. Regarding metabolites identification, in the text, metabolites are mentioned as putative metabolites to emphasize the identification confidence level in accordance to http://dx.doi.org/10.1007/s13361-016-1469-y; these authors proposed five identification levels based on type and quantity of evidence, which translates into confidence. Putative identification (metabolite identification confidence level 2) is indeed based on the use of orthogonal information like fragmentation data (which was performed in our study), yet validation is a must to confirm metabolite’s identity, so we agree with your comment about this need and, as it has been pointed out in the text, we are proposing to confirm the identity of metabolites in a future study along with proper experiments to make a quantitative study. We have clarified this intention in the text at the end of the discussion section.
Point 2. Confusing Results and Discrepancies:
- The results presented in the manuscript are somewhat perplexing. For instance, the direction of fold change for the two drugs in the MDA-MB-231 cell lines is largely opposite.
Response: It is due to both molecules (HO-AAVPA and VPA) have different physicochemical and structural properties, in which VPA is a short-chain fatty acid, while HO-AAVPA is an o-hydroxy-phenylamide derivative of VPA with better antiproliferative effects than VPA on MDA-MB-231 (https://pubmed.ncbi.nlm.nih.gov/27483122/) and lower toxic properties than VPA on fetus rats (https://pubmed.ncbi.nlm.nih.gov/32387339/). Then, we conclude that VPA and HO-AAVPA reach HDAC as reported https://pubmed.ncbi.nlm.nih.gov/32824279/ for HO-AAVPA and https://pubmed.ncbi.nlm.nih.gov/25782916/ for VPA. However, due to the mentioned differences in their physicochemical and biological responses, it has been shown in various in vitro and in vivo preclinical studies that this anticonvulsant drug significantly inhibits cancer cell proliferation by modulating multiple signaling pathways (https://pubmed.ncbi.nlm.nih.gov/34298623/ and https://pubmed.ncbi.nlm.nih.gov/36864445/); which explain their different metabolite profile found by LC-MS as reported for other HDAC inhibitors (hydroxamic and benzamides derivatives) (https://www.ncbi.nlm.nih.gov/pmc/articles/PMC8620738/).
- Furthermore, there is a substantial difference in the number of reported metabolites for MDA-MB-231 and MCF7 (as shown in Table 2 & 3), and this issue needs a more comprehensive explanation to help readers understand the findings better.
Response: We appreciate your observation about this subject. As you mentioned, although MCF-7 cells treated with HO-AAVPA showed the highest number of dysregulated metabolites, only 13 pathways were found to be dysregulated as opposed to the other experimental conditions (between 6 and 9 dysregulated pathways) and only 2 or 3 pathways were significantly impacted (according to figure 2) in all the experimental conditions and these are the most relevant insights, so even though the number of putative metabolites varies indeed, the number of significantly dysregulated pathways do not. We have clarified this issue in the text and thank you for bringing this situation to our attention.
Point 3. Inconsistencies in Impact Scores: I noticed a discrepancy in the impact scores for HO-AAVPA and VPA in MDA-MB-231 cells. Despite obtaining identical impact scores for some aspects in the metabolomics analysis, the direction of fold change for most identified metabolites in MDA-MB-231 differs between the two drugs. This requires further clarification and analysis to resolve the inconsistencies.
Response: The pathway enrichment analysis uses the hypergeometric test to calculate the pathway impact score. This score considers the proportion of metabolites that have been deregulated significantly, but it does not take into account the direction of fold change (https://doi.org/10.3390/biom13020244). This issue has been clarified in the text. HO-AAVPA and VPA show identical impact scores in some pathways, suggesting that both compounds affect the same pathways. However, it needs to be clarified whether these treatments have different effects on the regulation points within the metabolic pathway or if they have opposing effects on the same regulation points. Furthermore, it is important to note that metabolomic analysis only captures a static moment in time of a biological system. Various factors, such as production deficiencies, excess, or rapid metabolism, can cause metabolite changes. Further investigations and analyses are required to understand the metabolic pathways and cellular processes involved. These studies will help us to elucidate the mechanisms of action exerted by HO-AAVPA and VPA on these cells.
Point 4. Lack of Validation Methods and Clinical Relevance: The study would greatly benefit from the inclusion of validation methods using alternative techniques and patient-derived materials. This would help establish the clinical significance of the findings and improve the robustness of the conclusions.
Response: While clinical studies are quite valuable due to their likeliness to what can be found in patients, cell lines have their own value as biological models and show no restrictions found in patients’ biological material. Also, our approach to this problem is part of a well-established workflow (https://doi.org/10.1002/elps.201200605 and https://doi.org/10.3390/proteomes10030026), also used in proteomics, which consists in first, perform an untargeted analysis (hypothesis generator) and, once we have molecules of interest, a targeted analysis is performed (in a second study which is known as hypothesis-driven) that requires validation of metabolites’ identity and quantification to be sure that regulation changes observed in the first study are actually real. Also, part of the criteria for selecting both cell lines is due to their opposite prognosis and aggressiveness within the BC spectra, as stated in line 51.
Minor Issues:
Point 5. Introduction Simplification: I suggest simplifying the introduction by deleting lines 31-36 and starting with "Breast cancer (BC) currently has the highest incidence (considering 37 males and females) and..."
Response: Thank you, it has been re-written into the article text and labeled in yellow color.
Point 6. Improved Clarity of Methods: The description of the untargeted metabolomic study (Line 103-105) requires better clarity. Please rephrase this section to provide a more explicit explanation of the aims of the study, the "features" being referred to, and the utilization of fragmentation information for annotating corresponding putative metabolites.
Response: Thank you, it has been re-written into the article text and labeled in yellow color.
Point 7. In conclusion, while the topic of this manuscript is interesting and relevant, it is important to address the major issues concerning technique validation, result discrepancies, and the need for validation methods for clinical relevance. Moreover, the minor issues should be corrected to improve the overall clarity of the manuscript.
Response: Thank you, it has been re-written into the article text and labeled in yellow color.
Point 8. Comments on the Quality of English Language. While the topic of this research is intriguing and relevant, I must bring attention to a significant concern regarding the use of the English language throughout the manuscript. There were instances where I found it challenging to comprehend the intended message (e.g., line 103-105, line 91-98, etc.), which hindered the overall readability and clarity of this work. To improve the manuscript's quality and effectively convey the findings, I strongly recommend focusing on using simple and concise English language. This approach will help ensure that readers can easily understand and grasp the content without unnecessary complexities.
Response: We appreciate your recommendations. The English language has been revised and corrected.
Reviewer 2 Report (New Reviewer)
Summary:
Breast cancer is one of the leading causes of death in women worldwide. Treatment options are very limited for BC, especially at later stage. Developing alternative treatments are urgently needed. There are many subtypes in BC, which increases the difficulty of treatment due to the highly diversity of genetic and epigenetic background. Targeting epigenic pathways has been studies for years and has been reported to be a potential new hope. Compounds that inhibit HDAC activity, such as VPA (valproic acid), was shown to be a promising drug for cancer treatment. VPA was developed in treating epilepsy, bipolar disorder and schizophrenia. Thus, although the anti-proliferative effect of VPA seems promising, the major toxicity to normal cells and unbearable side effects are not neglectable. HO-AAVPA, a derivative from VPA, showed a promising low toxicity and high potency in BC cell models. Mechanisms for both compounds, VPA and HO-AAVPA have not been clarified. Main purpose in this manuscript is to decipher the mechanisms of VPA and HO-AAVPA-treated BC, specifically in metabolic pathways, which provide insights in developing new treatment strategies for BC. The authors found that the similarity of anti-proliferative effect from both compounds may be through similar mechanisms in MCF7 (ER+ model) and MDA-MB-231 (TNBC model) cell models; the discrepancy of the metabolite between the two compounds may explain the higher potency of HO-AAVPA v.s VPA.
Comments:
1. There is a large discrepancy compared to previous studies as mentioned in the result and discussion section; thus, providing a result for a reference compound would be a plus to validate the work here.
2. How does the modified metabolic pathways resulted from the two compounds related to their capability of inhibition on HDAC activity?
3. Please address the rationale of using MCF7 (ER+ model) and MDA-MB-231 (TNBC model) as the working models.
Author Response
Referee 2
Breast cancer is one of the leading causes of death in women worldwide. Treatment options are very limited for BC, especially at later stage. Developing alternative treatments are urgently needed. There are many subtypes in BC, which increases the difficulty of treatment due to the highly diversity of genetic and epigenetic background. Targeting epigenic pathways has been studies for years and has been reported to be a potential new hope. Compounds that inhibit HDAC activity, such as VPA (valproic acid), was shown to be a promising drug for cancer treatment. VPA was developed in treating epilepsy, bipolar disorder and schizophrenia. Thus, although the anti-proliferative effect of VPA seems promising, the major toxicity to normal cells and unbearable side effects are not neglectable. HO-AAVPA, a derivative from VPA, showed a promising low toxicity and high potency in BC cell models. Mechanisms for both compounds, VPA and HO-AAVPA have not been clarified. Main purpose in this manuscript is to decipher the mechanisms of VPA and HO-AAVPA-treated BC, specifically in metabolic pathways, which provide insights in developing new treatment strategies for BC. The authors found that the similarity of anti-proliferative effect from both compounds may be through similar mechanisms in MCF7 (ER+ model) and MDA-MB-231 (TNBC model) cell models; the discrepancy of the metabolite between the two compounds may explain the higher potency of HO-AAVPA v.s VPA.
Comments:
Point 1. There is a large discrepancy compared to previous studies as mentioned in the result and discussion section; thus, providing a result for a reference compound would be a plus to validate the work here.
Response: Thank you for your suggestion. Valproic acid (VPA) was included as a positive control. VPA is a well-known HDAC inhibitor but has not been characterized in depth for both cell lines (MCF-7 and MDA-MB-231). We have also compared our results with previous studies from within and outside our research group that differ from our experimental conditions, enriching the knowledge about VPA and getting insight into HO-AAVPA (a novel molecule whose effect has not been well characterized).
Point 2. How does the modified metabolic pathways resulted from the two compounds related to their capability of inhibition on HDAC activity?
Response: Numerous studies have shown that HDAC inhibitors are capable of regulating energy (https://doi.org/10.1210/me.2008-0179, https://doi.org/10.3389/fmolb.2021.634874, and https://doi.org/10.1016/j.bbrc.2018.11.103) and lipid metabolism in in vitro (https://doi.org/10.1186/1471-2164-9-507) and in vivo models (https://doi.org/10.3390/ijms222011224) by transcription-dependent and independent mechanisms. This is achieved by modifying the expression of enzymes that regulate the pathway or proteins involved in metabolite transportation.
Point 3. Please address the rationale of using MCF7 (ER+ model) and MDA-MB-231 (TNBC model) as the working models.
Response: Our work aims to demonstrate the impact that HO-AAVPA can have on both ER+ and TNBC models. We analyzed two cell lines with varying levels of aggressiveness and distinct biological targets, including HDACs and metabolomic profiles. Furthermore, numerous studies have been carried out on these cell lines, resulting in an extensive characterization of these cell lines (https://doi.org/10.7150/jca.18457).
Round 2
Reviewer 1 Report (New Reviewer)
Authors to consider the following amendments:
Title to amend to “Untargeted LC‒MS/MS Metabolomics Study of HO-AAVPA 2 and VPA on Breast Cancer Cell Lines”
Line 15-16: "Breast cancer (BC) is one of the biggest health problems worldwide. Its biochemical complexity is due to huge differences among molecular dysregulated pathways." Consider changing to "Breast cancer (BC) is one of the biggest health problems worldwide, characterized by intricate metabolic and biochemical complexities stemming from pronounced variations across dysregulated molecular pathways."
Line 17-18: "Then, it is required to continue searching for new treatment alternatives targeting epigenetic pathways such as histone deacetylases (HDACs). " Consider changing to "Thus, the pursuit of novel therapeutic avenues persists, notably focusing on epigenetic pathways such as histone deacetylases (HDACs)."
Line 19-20: "In this sense, a promissory compound, N-(2-hydroxyphenyl)-2- propylpentanamide (HO-AAVPA), derived from valproic acid (VPA) has been studied at the pre-clinical level." Consider changing to "The compound N-(2-hydroxyphenyl)-2-propylpentanamide (HO-AAVPA), a derivative of valproic acid (VPA), has emerged as a promising candidate warranting pre-clinical investigation."
Line 318-321: The differences observed between the studies mentioned above might be due to differences in sample preparation methods, the compound concentration used, the exposition time to the compound, storage, data acquisition methods (including the acquisition platform), etc.,and other factors, as has been noted in other publications [62–64].
- I believe instead of “exposition”, it should be “exposure”
Line 328: Results gathered through this study put some interesting insights that required further confirmation by a target approach.
- “Instead of “put some”. Consider substituting with “reveal”
Line 567-568: "Future studies must address complete chemical characterization to confirm the identity of the putative metabolites reported in this work." to change to "Future studies are required to confirm the identity of the putative metabolites reported in this work.
The quality is much better now with in this revised version.
Author Response
Point 1. Title to amend to “Untargeted LC‒MS/MS Metabolomics Study of HO-AAVPA 2 and VPA on Breast Cancer Cell Lines”
Response: Thank you for your recommendation. As per your suggestion, we have amended the title accordingly.
Point 2. Line 15-16: "Breast cancer (BC) is one of the biggest health problems worldwide. Its biochemical complexity is due to huge differences among molecular dysregulated pathways." Consider changing to "Breast cancer (BC) is one of the biggest health problems worldwide, characterized by intricate metabolic and biochemical complexities stemming from pronounced variations across dysregulated molecular pathways."
Response: At the author's suggestion, the text has been modified.
Point 3. Line 17-18: "Then, it is required to continue searching for new treatment alternatives targeting epigenetic pathways such as histone deacetylases (HDACs). " Consider changing to "Thus, the pursuit of novel therapeutic avenues persists, notably focusing on epigenetic pathways such as histone deacetylases (HDACs)."
Response: At the author's suggestion, the text has been modified.
Point 4. Line 19-20: "In this sense, a promissory compound, N-(2-hydroxyphenyl)-2- propylpentanamide (HO-AAVPA), derived from valproic acid (VPA) has been studied at the pre-clinical level." Consider changing to "The compound N-(2-hydroxyphenyl)-2-propylpentanamide (HO-AAVPA), a derivative of valproic acid (VPA), has emerged as a promising candidate warranting pre-clinical investigation."
Response: At the author's suggestion, the text has been modified.
Point 5. Line 318-321: The differences observed between the studies mentioned above might be due to differences in sample preparation methods, the compound concentration used, the exposition time to the compound, storage, data acquisition methods (including the acquisition platform), etc.,and other factors, as has been noted in other publications [62–64].
- I believe instead of “exposition”, it should be “exposure”
Response: At the author's suggestion, the text has been modified.
Point 6. Line 328: Results gathered through this study put some interesting insights that required further confirmation by a target approach.
- “Instead of “put some”. Consider substituting with “reveal”
Response: At the author's suggestion, the text has been modified.
Point 6. Line 567-568: "Future studies must address complete chemical characterization to confirm the identity of the putative metabolites reported in this work." to change to "Future studies are required to confirm the identity of the putative metabolites reported in this work.
Response: At the author's suggestion, the text has been modified.
Point 7. Comments on the Quality of English Language
The quality is much better now with in this revised version.
Response: The English text has been revised, and we thank the reviewer for their valuable suggestions that greatly improved its quality.
Reviewer 2 Report (New Reviewer)
The author clearly addressed the issues and the revised version is greatly improved. Overall, the presented data supports the conclusions. The study has a great potential. The observations are important for the community and therefore suggests for publication.
Author Response
Point 1. The author clearly addressed the issues and the revised version is greatly improved. Overall, the presented data supports the conclusions. The study has a great potential. The observations are important for the community and therefore suggests for publication.
Response: We express our gratitude for the reviewer's valuable time and insightful feedback on our manuscript. We acknowledge that their comments have undoubtedly contributed to the enhancement of the document.
This manuscript is a resubmission of an earlier submission. The following is a list of the peer review reports and author responses from that submission.
Round 1
Reviewer 1 Report
Major comments
In this work, the authors identified several metabolites dysregulated in the presence of HO-AAVPA and VPA using MDA-MB-231 and MCF7 cell lines. I found that this initial work is interesting but is missing the necessary validation experiment for at least two more dysregulated compounds related to the inhibitory effect of HO-AAVPA and VPA. Authors should provide precursor ion peaks with retention times and fragmentation patterns of compounds identified in this study and compare them with corresponding commercially available compounds. This will be directly prove that the identified compounds are not a false positive result.
Authors should also include a non-cancerous cell line (e.g. MCF10) as a control.
Minor comments
In the abstract, it is not appropriate to say that MCF7 is “non-triple negative breast cancer”. MCF7 is metastatic and MDA-MB-231 cell line is highly metastatic.
The main conclusion claimed by authors “
“The LC-MS untargeted metabolomic study allow for the simultaneous measuring of multiple metabolites and pathways identifying that both compounds share common effects on glycerophospholipid and sphingolipid metabolism on MCF-7 and MDA-MB-231 cell 24 lines.” could be interesting if it turns out that this is not the case with control non-cancerous cell line.
The concentration of VPA is very high (up to 36 mM). In addition to inhibitory effect on HDAC, this high concentration can affect different stress response on cells.
In Table 2 and 3 m/z values, fold change (not up/down) and the method used (HILIC or RF) should also be presented.
Figure 2 is not clear. What do the different circles represent? How is pathway impact calculated? what pathway?
In the line 314 the sentence “Sequentially 250, 350 and 250 μL of chloroform, water and chloroform were added and mixed…” is not clear.
The isolation window width should be maximum 2 amu (~ 4 amu is very high)
Author Response
Reviewers’ comments:
Major comments
In this work, the authors identified several metabolites dysregulated in the presence of HO-AAVPA and VPA using MDA-MB-231 and MCF7 cell lines. I found that this initial work is interesting but is missing the necessary validation experiment for at least two more dysregulated compounds related to the inhibitory effect of HO-AAVPA and VPA.
Response:. In this case, we are using valproic acid (VPA) as a positive control due to N-(2-hydroxyphenyl)-2-propylpentanamide (HO-AAVPA) is a novel compound derived from VPA, which is a HDAC inhibitor and has antiproliferative effects on breast cancer cell lines (https://www.sciencedirect.com/science/article/pii/S1319016422002936 and https://www.frontiersin.org/articles/10.3389/fcell.2022.1014798/full), as is describe into the work, we add additional phrases to support and clear it.
Authors should provide precursor ion peaks with retention times and fragmentation patterns of compounds identified in this study and compare them with corresponding commercially available compounds. This will be directly prove that the identified compounds are not a false positive result.
Response: There are enough replicates and statistical analyses (described at Methodology) to discard those negative signals, these LC-MS include the MS/MS analyses that guarantee the MW of the found molecules to be then submitted for (https://analyticalsciencejournals.onlinelibrary.wiley.com/doi/10.1002/mas.21562).
Authors should also include a non-cancerous cell line (e.g. MCF10) as a control.
Response: I The aim of this work was to study the metabolic profiles of luminal (MCF-7) and triple negative breast cancer (MDA-MB-231) cell lines in response to N-(2-hydroxyphenyl)-2-propylpentanamide (HO-AAVPA) to determine biochemical alterations and deepen the study of its effects at molecular level and its comparison against known HDAC inhibitor valproic acid (VPA). The antiproliferative effect of HO-AAVPA on non-cancerous cell line MCF-10A was reported previously (https://www.tandfonline.com/doi/full/10.1080/14756366.2016.1210138).
Minor comments
In the abstract, it is not appropriate to say that MCF7 is “non-triple negative breast cancer”. MCF7 is metastatic and MDA-MB-231 cell line is highly metastatic.
Response: We replaced “non-triple negative breast cancer (MCF-7)” by “luminal breast cancer,(MCF-7)”
The main conclusion claimed by authors “
“The LC-MS untargeted metabolomic study allow for the simultaneous measuring of multiple metabolites and pathways identifying that both compounds share common effects on glycerophospholipid and sphingolipid metabolism on MCF-7 and MDA-MB-231 cell 24 lines.” could be interesting if it turns out that this is not the case with control non-cancerous cell line.
Response: Thank you for your suggestion, due to our interest was comparing luminal breast cancer and triple negative breast cancer to identify the common and non-commun pathways under metabolomic investigations.
The concentration of VPA is very high (up to 36 mM). In addition to inhibitory effects on HDAC, this high concentration can affect different stress responses on cells.
Response: There were used drugs concentrations (IC15) according to the IC50 values (ref 45, 46) listed in Table 1. It is know elsewhere that the IC50 of VPA is lower than its derived tested here named HO-AAVOA (ref 31, 45, 46)
In Table 2 and 3 m/z values, fold change (not up/down) and the method used (HILIC or RF) should also be presented.
Response: Thank you for your suggestion, to avoid redundant information there were to include only the names and upregulate or downregulate according to fold values to group into two groups, the method used was HILIC mentioned in the methodology section.
Figure 2 is not clear. What do the different circles represent? How is pathway impact calculated? what pathway?
Response: The circles and their color represent the highest association with some affected intracellular pathways in presence of either VPA or HO-AAVOA to each corresponding cell used. The Table 4 lists the principal pathways affected from upper to lower disregulate metabolites.
In the line 314 the sentence “Sequentially 250, 350 and 250 μL of chloroform, water and chloroform were added and mixed…” is not clear.
Response: It was re-write into the article text.
The isolation window width should be maximum 2 amu (~ 4 amu is very high)
Response: For Agilent Q-TOF instrumentation, the isolation windows are narrow (≈1.3 amu), medium (≈4 amu), or large (≈9 amu), for that reason we used the medium (https://pubs.acs.org/doi/10.1021/acs.analchem.6b04358).
Reviewer 2 Report
A brief summary The aim of this study was not clearly explained. The authors have stated that the goal of the study is to evaluate the effect produced by compounds like HO-AAVPA and its comparison against known HDACi like VPA, a compound involved in the restoring of deregulated mechanisms in BC cells and to find new evidence about the metabolic pathways which, when restored, induced the activation of the mechanisms involved in the elimination of BC cells from different subtypes: LA, represented by MCF-7 and TNBC by MDA-MB-231.
The goal must be more specific.
Article: The hypothesis is not clearly stated as well as the contributions of the paper.
- Specific comments
- Line 30 – Incorrect and imprecise statement. According to the WHO definition, cancer is a large group of diseases that can start in almost any organ or tissue of the body when abnormal cells grow uncontrollably, go beyond their usual boundaries to invade adjoining parts of the body and/or spread to other organs.
- Lines 73-76– Missing reference for this statement
- Lines 79-80- Please be specific about the definition of metabolomics. What did the authors mean by the global view?
- Lines 80-87- This sentence is not clear. The aim of the study is not clearly explained.
- Line 94 - Why did the authors choose IC15 concentrations to study metabolomics?
- Lines 110-114 - The graphs should indicate which cell lines and which compounds they refer to.
- Lines 205-207 - The authors should treat cells with different concentrations of tested compounds in order to prove this statement.
- General questions to help guide your review report for research articles:
- Is the manuscript clear, relevant to the field and presented in a well-structured manner?
The Manuscript is not clear enough.
- Are the cited references mostly recent publications (within the last 5 years) and relevant? Does it include an excessive number of self-citations? Yes
- Is the manuscript scientifically sound and is the experimental design appropriate to test the hypothesis? The hypothesis is not clear.
- Are the manuscript’s results reproducible based on the details given in the methods section? Yes
- Are the figures/tables/images/schemes appropriate? Do they properly show the data? Are they easy to interpret and understand? Is the data interpreted appropriately and consistently throughout the manuscript? Please include details regarding the statistical analysis or data acquired from specific databases.
Figures are not easy to understand.
- Are the conclusions consistent with the evidence and arguments presented? It is not clear.
- Please evaluate the ethics statements and data availability statements to ensure they are adequate.
Extensive editing of English language required.
Author Response
Comments and Suggestions for Authors
A brief summary The aim of this study was not clearly explained. The authors have stated that the goal of the study is to evaluate the effect produced by compounds like HO-AAVPA and its comparison against known HDACi like VPA, a compound involved in the restoring of deregulated mechanisms in BC cells and to find new evidence about the metabolic pathways which, when restored, induced the activation of the mechanisms involved in the elimination of BC cells from different subtypes: LA, represented by MCF-7 and TNBC by MDA-MB-231.
The goal must be more specific.
Article: The hypothesis is not clearly stated as well as the contributions of the paper.
Response: Thank you for your suggestions, we have re-write the full document to clear the article content.
- Specific comments
- Line 30 – Incorrect and imprecise statement. According to the WHO definition, cancer is a large group of diseases that can start in almost any organ or tissue of the body when abnormal cells grow uncontrollably, go beyond their usual boundaries to invade adjoining parts of the body and/or spread to other organs.
- Response: Thank you for your suggestion, we have taken your suggested text to be included into the article.
- Lines 73-76– Missing reference for this statement
- Response: Thank you, the references have been included.
- Lines 79-80- Please be specific about the definition of metabolomics. What did the authors mean by the global view?
- Response: Thank you, there has been added a definition regarding to metabolomic.
- Lines 80-87- This sentence is not clear. The aim of the study is not clearly explained.
- Response:
- Line 94 - Why did the authors choose IC15 concentrations to study metabolomics?
- Response: Thank you, the phase was re-write to clear the main objective of the work.
- Lines 110-114 - The graphs should indicate which cell lines and which compounds they refer to.
- Response: At the Figure label there are letters (A-D) which mention each compound on the corresponding cell, it refers to antiproliferative effects, it has been included.
- Lines 205-207 - The authors should treat cells with different concentrations of tested compounds in order to prove this statement.
- Response: We agree to explore different concentrations, however, it is not recommendable to use from IC16 to IC100 values due to the number of cells not enough to see the metabolic behavior due to the low concentrations. The IC15 has been explored in other reports [https://pubmed.ncbi.nlm.nih.gov/28822947/].
- General questions to help guide your review report for research articles:
- Is the manuscript clear, relevant to the field and presented in a well-structured manner?
The Manuscript is not clear enough.
- Are the cited references mostly recent publications (within the last 5 years) and relevant? Does it include an excessive number of self-citations? Yes
- Is the manuscript scientifically sound and is the experimental design appropriate to test the hypothesis? The hypothesis is not clear.
- Are the manuscript’s results reproducible based on the details given in the methods section? Yes
- Are the figures/tables/images/schemes appropriate? Do they properly show the data? Are they easy to interpret and understand? Is the data interpreted appropriately and consistently throughout the manuscript? Please include details regarding the statistical analysis or data acquired from specific databases.
Figures are not easy to understand.
- Are the conclusions consistent with the evidence and arguments presented? It is not clear.
- Please evaluate the ethics statements and data availability statements to ensure they are adequate.
Round 2
Reviewer 1 Report
The authors did not provide any validation experiment
Author Response
The authors did not provide any validation experiment
Response: Thank you, it is a first approach following a previously strategy on cancer cells without treatment https://pubmed.ncbi.nlm.nih.gov/36292927/. In this case, we are using the same cell lines but in presence of VPA and its derivative (HO-AAVPA). And despite of the high probabilities of identifying metabolites due to the mass identify is from 3-4 digit numbers searched by specialized software (https://www.ncbi.nlm.nih.gov/pmc/articles/PMC8963255/) as were apply in this work (methods), we are mentioning the searched metabolites as "putative metabolites" due to for their full chemical characterization are needed their standards or perform metabolite isolation (https://www.sciencedirect.com/science/article/pii/S016599362030217X). We hope in the future we can develop these mentioned experimental assays, but we need to published this first contribution.